# In Utero Exposure to Δ9-Tetrahydrocannabinol Leads to Postnatal Catch-Up Growth and Dysmetabolism in the Adult Rat Liver

**DOI:** 10.3390/ijms22147502

**Published:** 2021-07-13

**Authors:** Shelby L. Oke, Kendrick Lee, Rosemary Papp, Steven R. Laviolette, Daniel B. Hardy

**Affiliations:** 1Department of Physiology and Pharmacology, Schulich School of Medicine and Dentistry, Western University, 1151 Richmond Street, London, ON N6A 5C1, Canada; soke2@uwo.ca (S.L.O.); klee843@uwo.ca (K.L.); rosiepapp@gmail.com (R.P.); 2The Children’s Health Research Institute, The Lawson Health Research Institute, London, ON N6A 5C1, Canada; 3Department of Anatomy and Cell Biology, Schulich School of Medicine and Dentistry, Western University, 1151 Richmond Street, London, ON N6A 5C1, Canada; steven.laviolette@schulich.uwo.ca; 4Department of Obstetrics and Gynaecology, Schulich School of Medicine and Dentistry, Western University, 1151 Richmond Street, London, ON N6A 5C1, Canada

**Keywords:** Δ9-tetrahydrocannabinol, intrauterine growth restriction, liver, metabolism, triglycerides, oxidative stress, mitochondria, miR-203a-3p, miR-29a/b/c

## Abstract

The rates of gestational cannabis use have increased despite limited evidence for its safety in fetal life. Recent animal studies demonstrate that prenatal exposure to Δ9-tetrahydrocannabinol (Δ9-THC, the psychoactive component of cannabis) promotes intrauterine growth restriction (IUGR), culminating in postnatal metabolic deficits. Given IUGR is associated with impaired hepatic function, we hypothesized that Δ9-THC offspring would exhibit hepatic dyslipidemia. Pregnant Wistar rat dams received daily injections of vehicular control or 3 mg/kg Δ9-THC i.p. from embryonic day (E) 6.5 through E22. Exposure to Δ9-THC decreased the liver to body weight ratio at birth, followed by catch-up growth by three weeks of age. At six months, Δ9-THC-exposed male offspring exhibited increased visceral adiposity and higher hepatic triglycerides. This was instigated by augmented expression of enzymes involved in triglyceride synthesis (ACCα, SCD, FABP1, and DGAT2) at three weeks. Furthermore, the expression of hepatic DGAT1/DGAT2 was sustained at six months, concomitant with mitochondrial dysfunction (i.e., elevated p66shc) and oxidative stress. Interestingly, decreases in miR-203a-3p and miR-29a/b/c, both implicated in dyslipidemia, were also observed in these Δ9-THC-exposed offspring. Collectively, these findings indicate that prenatal Δ9-THC exposure results in long-term dyslipidemia associated with enhanced hepatic lipogenesis. This is attributed by mitochondrial dysfunction and epigenetic mechanisms.

## 1. Introduction

Cannabis is the most commonly used recreational drug among individuals of reproductive age. In 2019, the Canadian Cannabis Survey found that approximately 21% of female adolescents and young adults self-reported daily or almost daily use of cannabis, while 40% reported monthly use [1]. The American College of Obstetricians and Gynecologists has further found that 2–5% of individuals use cannabis during pregnancy, and this was increased to 15–28% among young women who lived in urban settings and whom were socioeconomically disadvantaged [2]. Along with the recent legalization of recreational cannabis in Canada and select American states, the use of cannabis during pregnancy has become increasingly popular despite limited evidence of its safety. Many pregnant individuals use cannabis to diminish nausea, anxiety and depression, as they believe it to be a ‘safe’ and natural alternative to prescription medications [3,4,5]. Alarmingly, gestational exposure to cannabis can increase the risk for adverse neonatal outcomes, including low birth weight and preterm delivery [6,7,8]. However, these studies are confounded by that fact that cannabis users are more likely to concurrently abuse other drugs, such as alcohol and tobacco [9], highlighting the need for animal studies that focus on the effects of specific constituents of cannabis (i.e., Δ9-tetrahydrocannabinol [Δ9-THC] and cannabidiol [CBD]) on short- and long-term maternal-fetal outcomes [9].

Cannabis is composed of several distinct compounds that stimulate the endocannabinoid system via the interaction with cannabinoid receptor type 1 (CB1R) and CB2R. These receptors are highly expressed throughout the central nervous system (CNS). Therefore, their activation by cannabinoids has a profound effect on mood, pain, memory and appetite [10]. At the same time, CB1R and CB2R are localized in peripheral metabolic tissues, suggesting that the endocannabinoid system has additional roles outside of neurocognitive function [11,12,13,14]. Many cannabinoids are also lipophilic and able to cross the placental barrier, thereby entering fetal circulation and allowing for activation of CB1R and CB2R [15]. While the activation of the endocannabinoid system is essential for the maintenance of pregnancy, the presence of exogenous cannabinoids can also directly interfere with fetal growth and development through the disruption of endocannabinoid signaling. Gestational exposure to Δ9-THC, the principal psychoactive component of cannabis, has been demonstrated to promote placental insufficiency and symmetrical intrauterine growth restriction (IUGR) in rodents, characterized by low birth weight and reduced neonatal organ weight [16,17,18,19,20,21]. Notably, the liver-to-body weight ratio is reduced in Δ9-THC offspring, followed by rapid catch-up growth in the first three weeks of life [21]. While the developmental origins of health and disease (DOHaD) postulates that there is an inverse relationship between birth weight and metabolic health, the role of gestational Δ9-THC exposure on postnatal hepatic function has not yet been investigated.

The liver has a critical role in controlling lipid metabolism, which involves the synthesis and degradation of structural and functional lipid molecules. Naturally, impaired liver function leads to dyslipidemia, whereby triglycerides and cholesterol become elevated in the liver and plasma. Dyslipidemia is a well-characterized attribute of the metabolic syndrome that often accompanies obesity. Therefore, individuals with either of these conditions exhibit an increased risk for developing diabetes and cardiovascular disease [22]. Moreover, epidemiological studies have identified that low birth weight individuals with decreased liver-to-body weight ratio are more likely to be obese and have non-alcoholic fatty liver disease (NAFLD) during childhood and adult life [23,24,25,26,27,28,29]. Similar trends have been found in studies of rodent IUGR offspring. We previously shown that maternal nicotine exposure leads to elevated hepatic and circulating triglycerides in adult male offspring, while protein-restricted adult males have increased hepatic and circulating cholesterol [30,31]. Hepatic hyperlipidemia often occurs due to divergent synthesis and metabolism of free fatty acids, which can be of dietary, circulating or de novo origin [32]. In particular, de novo lipogenesis is a highly regulated, multi-step process that involves the esterification of fatty acids to triglycerides through the action of numerous enzymes. Initially, acetyl-coA carboxylase (ACCα) catalyzes the carboxylation of acetyl-coA to malonyl-coA, which acts as a substrate for fatty acid synthase (FAS) in generating saturated fatty acids [32,33]. Stearoyl-coA desaturase-1 (SCD-1) then converts these saturated fatty acids into monounsaturated fatty acids (MUFAs), which undergo various elongation reactions to generate the long chain fatty acids that feed into triglyceride synthesis pathways [34]. The cytosolic transport of these long chain fatty acids is mediated by various fatty acid binding proteins (FABPs), which are highly expressed in tissues that control lipid metabolism. Importantly, the diacylglycerol acyltransferase (DGAT) enzyme catalyzes the terminal step of triglyceride synthesis [35,36]. Studies of the growth-restricted liver demonstrate that many of these enzymes are particularly sensitive to developmental reprogramming, as their abundance and activity levels change with exposure to a variety of gestational insults leading to long-term dyslipidemia [30,37,38,39]. That said, the underlying mechanisms linking a poor in utero environment to these metabolic deficits remain elusive.

Oxidative stress and mitochondrial dysfunction occur with many metabolic pathologies, including hepatic dyslipidemia. Many studies have shown that growth-restricted offspring exhibit oxidative stress and impaired mitochondrial metabolism in the liver [40,41,42,43,44,45]. Therefore, it is possible that mitochondrial dysfunction precedes dyslipidemia in IUGR offspring. This may be mediated, in part, by the adaptor protein p66Shc, which is known to accelerate the mitochondrial production of ROS [46]. Human and animal studies indicate that the p66Shc-induced oxidative signal is involved in the accumulation of intracellular lipids, leading to obesity and NAFLD [47,48]. Not surprisingly, growth-restricted offspring display increased levels of p66Shc in the liver, kidney, and pancreas at birth and in postnatal life [45,49,50]. Furthermore, using a rodent model of maternal protein restriction, we have also shown that adult male offspring have increased hepatic protein abundance of p66Shc following postnatal catch-up growth [45]. These same offspring also have aberrant expression of various microRNAs (miRs) in the liver, so it is possible that the expression of genes involved in de novo lipogenesis and mitochondrial function may be regulated via epigenetic mechanisms [51]. Based on the fact that gestational exposure to Δ9-THC leads to compromised liver growth, the present study investigates the effects of Δ9-THC on hepatic lipid metabolism in the exposed offspring. Given the role of mitochondria in metabolic disease and the growth-restricted liver, we hypothesized that Δ9-THC-exposed offspring would exhibit oxidative stress and mitochondrial dysfunction, along with aberrant expression of miRs that are known to result in dysmetabolism.

## 2. Results

### 2.1. Gestational Exposure to Δ9-THC Leads to Hepatic Catch-Up Growth by Three Weeks of Age

At birth (i.e., PND 1), Δ9-THC-exposed offspring exhibited decreased liver to body weight ratio compared to control offspring (Table 1; *p* < 0.05). It should be noted that in this same cohort of vehicle and Δ9-THC offspring, we have published that Δ9-THC exposure during pregnancy did not lead to changes in maternal food intake, maternal weight gain, litter size or gestational length [21]. By three weeks of age, these offspring had exhibited hepatic catch-up growth, as there were no significant differences between liver to-body-weight ratios of Δ9-THC-exposed offspring and control offspring (Table 1). This is consistent with other rodent models of IUGR, including maternal nicotine exposure and maternal protein restriction, whereby growth-restricted offspring also undergo catch-up growth by three weeks [31,52]. At three weeks of age, there were also no differences in liver-to-body weight ratios between male and female offspring of either treatment group (Table 1), indicating that there were no sex-specific effects of gestational Δ9-THC with respect to hepatic catch-up growth. At six months of age, liver to body weight ratio remained equal between both groups and sexes (Table 1). It should also be noted that gestational exposure to Δ9-THC did not significantly affect postnatal food intake (PND50–60) in either males (vehicle, 19.3 ± 1.5 g food/day/offspring; Δ9-THC, 15.12 ± 2.1 g food/day/offspring) or females (vehicle, 17.7 ± 1.6 g food/day/offspring; Δ9-THC, 20.03 ± 1.9 g food/day/offspring.

### 2.2. Adult Δ9-THC-Exposed Offspring Exhibit Elevated Visceral Adiposity and Hepatic Dyslipidemia Following Gestational Exposure to Δ9-THC

As mentioned previously, adult IUGR offspring are vulnerable to the development of obesity and hepatic pathologies involved in lipid storage and metabolism. Given that Δ9-THC induces symmetrical IUGR [21], we were interested in comparing the levels of visceral adipose tissue and hepatic lipids between control and Δ9-THC-exposed offspring during adult life. At six months, offspring exposed to gestational Δ9-THC exhibited increased visceral adipose to body weight ratio (Table 1; *p* < 0.05), suggesting dyslipidemia in these animals.

That said, this was true only when examining both sexes together, as there were differences in visceral adipose to body weight ratio when examining each sex individually. To explore this further, we measured hepatic and circulating triglyceride and cholesterol levels of each sex for both groups at six months of age. While both hepatic and circulating triglyceride and cholesterol levels of Δ9-THC-exposed female offspring remained unchanged at six months (Figure 1A–D), hepatic triglycerides were elevated in Δ9-THC-exposed males in comparison to control males and all females (Figure 1A; *p* < 0.05). Interestingly, hepatic cholesterol was unaltered in six-month old male offspring exposed to gestational Δ9-THC, as were circulating triglycerides and cholesterol (Figure 1B–D). Since the differences in lipid content were observed exclusively in livers of male offspring, we further quantified hepatic triglycerides and cholesterol in male offspring at an earlier timepoint. At three weeks of age, hepatic triglycerides and cholesterol were not significantly altered between control and Δ9-THC-exposed male offspring (Figure 1E,F).

### 2.3. Elevated Hepatic Triglyceride Levels Coincide with Increased Diglycerol Acyltransferase and p66Shc Protein Levels in the Livers of Adult Male Δ9-THC-Exposed Offspring

To gain insight on possible mechanisms involved in the elevation of hepatic triglycerides among adult male Δ9-THC-exposed offspring, we performed western immunoblotting for enzymes involved in the triglyceride synthesis pathway. Δ9-THC-exposed males exhibited increased DGAT1 (Figure 2F; *p* < 0.05) and DGAT2 (Figure 2G; *p* < 0.05) in the liver at six months of age, while there were no significant differences in hepatic ACCα, FAS, SCD, or FABP1 (Figure 2B–E). Adult female Δ9-THC-exposed offspring also demonstrated a significant increase in DGAT2 (Figure 2N; *p* < 0.05) relative to female controls, while all other enzymes involved in triglyceride synthesis were unchanged (Figure 2I–M). As p66Shc is linked to the accumulation of intracellular lipids [47], we further quantified the relative abundance of p66Shc protein in control and Δ9-THC-exposed offspring. Adult male offspring exposed to gestational Δ9-THC demonstrated increased hepatic p66Shc protein abundance at 6 months compared to control male offspring (Figure 2H; *p* < 0.05), while p66Shc was unchanged in adult female offspring (Figure 2O). Again, because changes were observed in hepatic lipid content exclusively in adult male offspring, we further examined the expression of these enzymes in males at three weeks of age. At three weeks, Δ9-THC-exposed male offspring had increased protein abundance of DGAT2 (Figure 3G; *p* < 0.001) and FABP1 (Figure 3E; *p* < 0.01). However, protein levels of ACCα, FAS, SCD, DGAT1 and p66Shc were unchanged in male offspring at three weeks of age (Figure 3B–D,F,H).

### 2.4. Gestational Exposure to Δ9-THC Does Not Affect Protein Levels of Enzymes Involved in Aerobic Metabolism at Six Months of Age

Given that male offspring exhibit altered lipid metabolism and increased p66Shc, which are both implicated in mitochondrial dysfunction [53,54], we then assessed protein levels of enzymes involved in aerobic metabolism. At six months, the abundance of mitochondrial transcription factor A (TFAM), a protein that is essential in activating mitochondrial genome transcription, was unaltered in both male and female offspring exposed to gestational Δ9-THC (Figure 4B,F). Furthermore, male offspring exhibited no changes in the ratio of phosphorylated pyruvate dehydrogenase at serine residue 232 (p-PDH[Ser232]) to the total PDH, lactate dehydrogenase subunit A (LDHa), or citrate synthase (Figure 4C–E). Interestingly, six-month female offspring exposed to gestational Δ9-THC displayed a decrease in the ratio of p-PDH[Ser232] to total PDH (Figure 4G), while protein levels of LDHa and citrate synthase were unchanged (Figure 4H,I). However, at three weeks of age, male Δ9-THC offspring exhibited increased levels of TFAM (Figure 5B; *p* < 0.01) and LDHa (Figure 5D; *p* < 0.01), while the ratio of p-PDH[Ser232] to total PDH and levels of citrate synthase were unchanged (Figure 5C,E).

### 2.5. Gestational Exposure to Δ9-THC Leads to Altered Hepatic Protein Levels of Superoxide Dismutase 1 in Adult Male Offspring

Given that the increased p66Shc is also associated with oxidative stress [46], we further analyzed the abundance of antioxidant enzymes that are critical in combatting the damaging effects of ROS. Male offspring demonstrated decreased protein levels of SOD1 at six months of age (Figure 6C; *p* < 0.05), while catalase and SOD2 remained unchanged at this time point (Figure 6B,D). Lipid peroxidation was also unaffected at six months of age in male offspring exposed to Δ9-THC, as indicated by levels of 4-hydroxynonenol (4HNE; Figure 6E). At three weeks, male Δ9-THC-exposed offspring did not exhibit any changes in catalase, superoxide dismutase (SOD) 1 or SOD2 protein levels (Figure 6F–H). However, hepatic lipid peroxidation was increased as indicated by elevated levels of 4HNE (Figure 6I; *p* < 0.05).

### 2.6. Mitochondrial Electron Transport Chain Complexes Are Increased in Male Offspring at Six Months and Three Weeks of Age Following Gestational Exposure to Δ9-THC

The mitochondrial electron transport chain is known to have significant contribution to the production of ROS, particularly through the activity of complexes I and III [55].

Given we observed changes in antioxidant and mitochondrial proteins in Δ9-THC offspring, we next quantified protein levels of each subunit of the electron transport chain. At six months, male offspring exposed to gestational Δ9-THC exhibited increased abundance of complexes I, III, and V (Figure 7B,D,F; *p* < 0.05), while levels of complexes II and IV were unchanged (Figure 7C,E). While adult female offspring demonstrated an increase in the protein abundance of complex I (Figure 7G; *p* < 0.01), the expression of all other subunits remained unchanged (Figure 7H–K). At three weeks, male Δ9-THC-exposed offspring also demonstrated increased levels of complexes I and III (Figure 8B,D; *p* < 0.05), while all other complexes were unchanged (Figure 8C,E,F).

### 2.7. Adult Male Offspring Exposed to Gestational Δ9-THC Exhibit Decreased Hepatic Transcript Levels of miR-203a-3p and miR-29a/b/c

Various miRs are known to be dysregulated with metabolic disease, and we have previously shown that IUGR offspring with postnatal catch-up growth exhibit altered expression of miRs, specifically, miR-29a/b/c [51]. To better understand the potential mechanisms behind the increase in p66Shc protein levels among adult male Δ9-THC-exposed livers, we first investigated whether hepatic transcript abundance of miR-203a-3p, which silences p66shc [56], was altered in these same offspring. As expected, transcript abundance of miR-203a-3p was significantly decreased in the livers of adult male offspring with gestational exposure to Δ9-THC (Figure 9A; *p* < 0.001). Furthermore, livers from three-week old male Δ9-THC-exposed offspring displayed a trending increase in miR-203a-3p transcript abundance, but this change was not significant (Figure 9E). Similar to miR-203a-3p, hepatic transcript abundances of miR-29a/b/c was also decreased in six-month old male offspring exposed to gestational Δ9-THC (Figure 9B–D), while the expression of each isoform was unchanged at three weeks of age (Figure 9F–H).

## 3. Discussion

Impairments in the hepatic lipogenic pathway promote excessive production and storage of intracellular triglycerides, leading to the development of obesity and the metabolic syndrome [22]. In the current study, we demonstrate that gestational exposure to Δ9-THC leads to increased adipose to bodyweight ratio at six months of age, along with elevated hepatic triglyceride levels in adult male offspring. The observed dyslipidemia coincides with increased hepatic expression of lipogenic enzymes at three weeks and six months of age, culminating in accelerated de novo lipogenesis during adult life. Our results suggest that these changes in lipid metabolism occur in a sex-specific manner, as hepatic triglyceride levels were unchanged in adult female offspring exposed to gestational Δ9-THC. Livers taken from three-week old Δ9-THC-exposed male offspring also showed increased 4HNE abundance, suggesting that hepatic lipid peroxidation and oxidative stress first occurs in early life. Furthermore, oxidative stress and mitochondrial dysfunction appear to persist into adulthood when hepatic triglycerides are increased. Last, our study provides insight into the epigenetic mechanisms that may underly hepatic dyslipidemia in growth-restricted offspring, as adult male offspring exposed to gestational Δ9-THC exhibited decreased expression of miR-203a-3p and miR-29a/b/c, all involved in mitochondrial homeostasis in the liver.

As mentioned previously, there is an inverse relationship between birth weight and long-term metabolic health. We have reported that gestational exposure to Δ9-THC leads to symmetrical IUGR in rodent offspring, followed by whole body and hepatic catch-up growth by three weeks of age [21]. The occurrence of catch-up growth, or rapid postnatal weight gain, is believed to further exacerbate the risk for metabolic diseases, such as obesity, as seen in birth cohorts studying growth-restricted infants from South Africa, Brazil, and the Unites States [57,58,59]. Here we show that Δ9-THC-exposed offspring display increased visceral adipose to body weight ratio at six months of age, indicating the development of obesity in adult life. This is consistent with our previous studies of rodent offspring born from models of maternal nicotine exposure and maternal protein restriction, whereby visceral obesity was observed in adult offspring following postnatal catch-up growth [60,61]. Clinically, obese individuals exhibit hypertriglyceridemia in the liver and plasma, often culminating in hepatic steatosis and later NAFLD [62]. In our model, gestational exposure to Δ9-THC led to an increase in hepatic triglycerides exclusively in male offspring at six months of age. Interestingly, there were no differences in the levels of circulating triglycerides among male offspring, while circulating and hepatic cholesterol levels of both sexes were also unaffected. We have previously reported that female Δ9-THC exposed offspring exposed to gestational Δ9-THC do not exhibit differences in circulating estrogen or testosterone relative to control female offspring, so this protective effect cannot be attributed to changes in the circulating levels of androgenic hormones [63]. However, the differences in steroid receptor (i.e., androgen receptor/estrogen receptor) signaling in the liver might be involved [64,65]. Overall, there is great evidence to suggest that men are more susceptible than women to the development of hepatic steatosis and NAFLD [66,67,68,69,70]. Studies of healthy men and women demonstrate that men exhibit higher rates of de novo lipogenesis and decreased dietary fatty acid oxidation; therefore, the sexual dimorphism observed among Δ9-THC-exposed offspring may be attributed to changes in lipogenesis and lipolysis [71,72].

Given the sex-specific differences in hepatic triglyceride content of Δ9-THC-exposed offspring, we next investigated the relative protein abundance of lipogenic enzymes among male and female adult offspring. At six months of age, male offspring exposed to gestational Δ9-THC exclusively displayed increased levels of both DGAT1 and DGAT2, along with an increase in the mitochondrial adaptor protein p66Shc. Deletion of DGAT and p66Shc are both independently associated with reduced intracellular accumulation of triglycerides [47,73]. Therefore, the upregulation of both DGAT1/2 and p66Shc in the current study likely mediates the increase in hepatic triglycerides of male Δ9-THC exposed offspring. While adult female Δ9-THC exposed offspring demonstrated increases in DGAT2, this may not be sufficient in promoting increased de novo lipogenesis in the liver. Instead, a synergistic interaction of multiple enzymes is necessary to elicit changes in lipid metabolism. Since we observed lipogenic changes in male offspring exposed to gestational Δ9-THC in adulthood, we further assessed whether this occurred earlier (e.g., three weeks) coinciding with the completion of hepatic catch-up growth. Δ9-THC exposed male offspring exhibited increased abundance of ACCα, SCD, FABP1 and DGAT2 at three weeks of age, suggesting that catch-up growth may instigate de novo lipogenesis. Incidentally, recent studies have determined that FABP1 is an important regulator of Δ9-THC hepatic transport and biotransformation [74,75]. Many hepatic pathophysiologies, including NAFLD, are associated with increased expression of FABP1, while its knockdown reduces hepatic triglyceride accumulation and lipid peroxidation [76]. This is consistent with our current study, as three-week old male offspring displayed increased abundance of 4HNE (i.e., a marker of lipid peroxidation) concomitant with elevated FABP1. Since FABP1 is involved in Δ9-THC metabolism, it is possible that these offspring have altered expression of FABP1 in response to exposure to Δ9-THC in utero. Mice with knockdown of hepatic FABP1 also exhibit decreased expression of DGAT1 and DGAT2 in the liver [76]; therefore, early elevation of FABP1 in Δ9-THC-exposed male offspring could be involved in the upregulation of DGAT and de novo lipogenesis later in life.

Oxidative stress and mitochondrial dysfunction are highly prevalent among metabolic diseases involving the liver. Elevated reactive oxygen species (ROS) and mitochondrial abnormalities are implicated in the pathogenesis of NAFLD. However, the mechanisms by which this occurs are not completely understood [77,78,79,80]. In vitro studies have established that ROS are detrimental to hepatic lipid metabolism and mitochondrial function, as treatment of HepG2 cells and primary hepatocytes with hydrogen peroxide promotes the accumulation of triglycerides and cholesterol [81]. Additionally, lipid-induced elevation of ROS has been shown to impair mitochondrial function in the human hepatoblastoma C3A cell line, along with increased gluconeogenesis and ketogenesis [82]. Numerous *animal* models have also found that IUGR offspring exhibit indices of hepatic oxidative stress (e.g., increased lipid peroxidation and ROS; altered expression and activity of antioxidant enzymes) and impaired mitochondrial function (e.g., aberrant expression and activity of pyruvate dehydrogenase, citrate synthase, and complexes of the electron transport chain; disrupted ATP synthesis) during neonatal and adult life [40,41,42,43,45]. In particular, we have shown that hepatic p66Shc is elevated in adult male offspring subject to maternal protein restriction attributed to postnatal catch-up growth [45]. P66Shc is a key regulator of cellular redox state, lifespan, and mitochondrial metabolism, as it interacts with cytochrome C under conditions of cell stress to stimulate the production of ROS [46]. This mitochondrial-induced oxidative stress often leads to apoptosis, cellular senescence and compromised aerobic metabolism [46,83,84,85], making p66Shc an attractive target in mitigating metabolic disease. Again, studies have also postulated that p66Shc is involved in the accumulation of intracellular lipids [47]. Here, we found that adult male offspring exposed to gestational Δ9-THC have increased protein abundance of p66Shc in the liver, concomitant with decreased abundance of SOD1. While we did not directly measure hepatic ROS, these results suggest that oxidative stress occurs in the liver following exposure to gestational Δ9-THC and catch-up growth. To investigate this further, we assessed the levels of proteins involved in mitochondrial function and aerobic metabolism (i.e., TFAM, p-PDH[Ser232], LDHa, citrate synthase, and complexes of the electron transport chain). While the expression of TFAM, p-PDH[Ser232], LDHa and citrate synthase was unchanged in adult male offspring, they did have increased amounts of complexes I, III and V. This is noteworthy as increases in complexes I and III are associated with elevated superoxide production via increased flow of electrons through the subunits of these enzymes [55]. These data, in combination with our p66Shc and SOD1 findings, further support the idea that these offspring exhibit hepatic oxidative stress and altered oxidative phosphorylation. Complexes I and III also increased in three-week old male offspring, suggesting that oxidative stress occurs in early life following Δ9-THC exposure and catch-up growth.

To date, little is known about the epigenetic mechanisms underlying metabolic diseases in IUGR offspring. One such mechanism is miRs, which are small, non-coding RNA molecules that silence target genes through mRNA degradation or repressed protein translation. Numerous miRs have been demonstrated to have aberrant postnatal expression in growth-restricted offspring, leading to cellular stress that precedes impaired function of metabolic organs [51,86,87,88]. Recent studies found that the translation of p66Shc protein is directly inhibited by miR-203a-3p, leading to the attenuation of liver injury and fibrosis in mice [56]. Here, we observed robust decreases in the expression of miR-203a-3p in the livers of male offspring exposed to gestational Δ9-THC, but only at six months of age. This is consistent with the observed increase in p66Shc protein levels at this time point, indicating that miR-203a-3p may be implicated in the development of oxidative stress and de novo lipogenesis in these offspring. We further quantified the expression of miR-29, as we have found this miR family to be altered in the livers of adult male protein-restricted offspring following catch-up growth [51]. Similar to our findings of miR-203a-3p, adult male Δ9-THC-exposed offspring exhibited decreased hepatic expression of all three isoforms of miR-29 at six months of age. This is of great interest as miR-29 expression is downregulated in patients with advanced liver fibrosis and cirrhosis [89]. Conversely, when present at high levels, miR-29 can alleviate hepatocellular steatohepatitis, fibrosis and cirrhosis in mice [89,90,91]. Collectively, given that each of these miRs were down-regulated exclusively during adult life, it is conceivable that early catch-up growth of the liver culminates in long-term oxidative stress and impaired hepatic lipid metabolism through epigenetic regulation of gene expression. Future in vitro studies are warranted to investigate this relationship further to reveal additional metabolic targets that become dysregulated with altered expression of miR-203a-3p and miR-29a/b/c in the liver.

In summary, this study demonstrates for the first time that gestational exposure to Δ9-THC leads to hepatic dyslipidemia in adult male offspring. We postulate that this occurs as a result of accelerated triglyceride synthesis (i.e., de novo lipogenesis) in the liver, as multiple lipogenic enzymes are elevated at both three weeks and six months of age. Given that FABP1 is involved in the metabolism and transport of Δ9-THC, it is possible that gestational exposure to Δ9-THC in combination with catch-up growth leads to early elevation of FABP1 in the livers of male offspring. This may further contribute to the observed increase in lipid peroxidation and lipogenic enzyme expression in male offspring at three weeks of age. Hepatic lipid overload is known to further increase oxidative stress and mitochondrial dysfunction. Therefore, this process may occur in a cyclical manner as indicated in Figure 10. Altered expression of miRs may also contribute to these molecular changes, as we have indicated that miR-203a-3p and miR-29a/b/c are downregulated in response to Δ9-THC exposure and catch-up growth. Additional long-term studies will be important in determining if this later culminates in hepatic pathologies such as NAFLD and cirrhosis of the liver. Given that Δ9-THC interacts with both the CB1 and CB2 receptors of the endocannabinoid system, it is also possible that differences in the expression of these receptors may exist upstream of the observed physiological and molecular effects. That said, the current study is somewhat limited in that we did not examine the expression of either CB1R or CB2R. While much remains unknown about the role of prenatal cannabinoids on offspring outcomes, it is clear that long-term metabolic health becomes compromised as a result. We have previously shown that exposure to Δ9-THC specifically has effect on glucose tolerance and cardiovascular function [63,92], and our current study demonstrates that it further contributes to dyslipidemia. Overall, these data provide great insight into the effects of gestational Δ9-THC exposure on the development and function of the liver, as well as the fundamental molecular mechanisms that underlie the metabolic dysfunction of growth-restricted offspring.

## 4. Materials and Methods

### 4.1. Animals and Experimental Handling

All procedures were performed according to guidelines set by the Canadian Council of Animal Care, and the animal use protocol was approved by the Animal Care Committee at The University of Western Ontario (AUP #2019-126, January 2019). All investigators understood and followed the ethical principles outlined by Grundy [93], and the study design was informed by ARRIVE guidelines [94]. Pregnant female Wistar rats were purchased from Charles River (La Salle, St. Constant, QC, Canada). Dams arrived at the animal care facility at gestational day 3 (GD3) and were left to acclimatize to environmental conditions for three days. All animals were maintained at 22 °C on a 12:12 h light-dark cycle in the animal care facility, while food and water were provided ad libitum for the entire duration of the experimental protocol. Dams were randomly assigned to receive daily intraperitoneal (*i.p.*) injection of either vehicular control (1:18 cremophor:saline) or 3 mg/kg Δ9-THC (Sigma-Aldrich, St. Louis, MO, USA) from E6.5 to E22 (n = 14, where litter is the statistical unit) as previously performed [21]. This dose of Δ9-THC was selected as it results in rodent plasma concentrations (8.6–12.4 ng/mL) that are reflective of those found in human recreational cannabis smokers (using 6% Δ9-THC) 0–22 h post-inhalation (13–63 ng/mL), as well as in the aborted fetal tissues of pregnant cannabis users (4–287 ng/mL) [95,96,97]. This dose and method of delivery has also been demonstrated to have no impact on maternal or litter outcomes in rats [21,98,99]. An oral route of administration was not chosen as to avoid the poor bioavailability and slowed adsorption of Δ9-THC when ingested with food, and as edibles are the least popular route of administration of cannabis among pregnant women [4,100]. Injections were initiated at E6.5 because Δ9-THC can interfere with implantation of the blastocyst and induce spontaneous abortion [100].

Maternal food intake and body weight were monitored daily throughout the entire gestational period. Dams were allowed to deliver normally, and all pups were weighed at birth. Litters were randomly culled to eight offspring per litter to ensure uniformity of litter size between treatment groups. As previously reported, we and others have found the selected dose of Δ9-THC has no impact on maternal or litter outcomes [20,21,98,99]. Liver weights of culled offspring were recorded and compared to body weight as a measure of fetal growth restriction and postnatal catch-up growth [21]. The food intake of these offspring was monitored by measuring their daily food consumption from PND50-60, as previously published [51]. The remaining offspring were fasted for 24 h before being euthanized via i.p. injection of 100 mg/kg pentobarbital at either postnatal day (PND) 21 or six months of age (n = 8 males/8 females per group), followed by necropsy to examine the effects of Δ9-THC on metabolic and molecular outcomes. The right medial hepatic lobe was collected and immediately flash-frozen in liquid nitrogen, followed by storage at −80 °C until further use. Visceral fat was also weighed and compared to body weight at six months as a measure of obesity. Blood was also collected, centrifuged, and stored at −80 °C.

### 4.2. Hepatic Triglyceride Measurements

Circulating and hepatic triglyceride and cholesterol measurements were detected using the Cobas^®^ Mira S analyzer as previously published [30]. For triglyceride measurements, triglycerides were hydrolyzed by lipoprotein lipase to glycerol and fatty acids. Glycerol was then phosphorylated to glycerol-3-phosphate by ATP in a reaction catalyzed by glycerol kinase (GK). The oxidation of glycerol-3-phosphate was catalyzed by glycerol phosphate oxidase (GPO) to form dihydroxyacetone phosphate and hydrogen peroxide (H_2_O_2_). In the presence of peroxidase, H_2_O_2_ alters the oxidative coupling of 4-chlorophenol and 4-aminophenazone to form a red-colored quinoneimine dye, which was measured at 512 nm. The increase in absorbance is directly proportional to the concentration of triglycerides in the sample. For cholesterol measurements, cholesterol esterase cleaved cholesterol esters, which then were converted to choleste-4-en-3-one and H_2_O_2_ by cholesterol oxidase. Cholesterol levels were quantified using a colorimetric assay that measured the breakdown of H_2_O_2_ via the Trinder reaction as previously described [31].

### 4.3. RNA Isolation and Quantitative Real-Time PCR Analysis

MicroRNAs were isolated from frozen liver samples using a miRNeasy kit (QIAGEN Inc., Toronto, ON, Canada), followed by spectrophotometric analysis with a Nanodrop 2000. 0.5 µg of each miRNA sample was then reverse transcribed into cDNA using a miScript II RT kit (QIAGEN Canada) and stored at −20 °C. Forward sequence primers for miR-203a-3p, miR-29a, miR-29b, and miR-29c were purchased (QIAGEN Canada), while a universal reverse sequence primer for miRNAs was used as part of a miScript SYBR green PCR kit (QIAGEN Canada). MiRNA transcripts were amplified via quantitative real-time PCR (qRT-PCR) using the Bio-Rad CFX384 Real Time System, using cycling conditions as previously published [51]. Cq values obtained for miRs of interest were normalized to that of a miRNA standard control (Ctrl_miRTC_1; QIAGEN Canada), and relative transcript abundance was calculated for each target using the comparative ΔCt method.

### 4.4. Protein Extraction and Western Immunoblot

Whole-cell protein lysates were isolated using the protocol, previously described by Barra et al. [52] Prior to western immunoblotting, loading samples were prepared using sample lysates, NuPAGE reducing agent (10×; Invitrogen, Waltham, MA, USA), NuPAGE LDS sample buffer (4X; Invitrogen), and deionized water. Loading samples were heated at 70 °C to denature proteins, while a separate set of loading mixes were kept unheated for mitochondrial OXPHOS immunoblots. Samples were loaded into wells of 4–12% Bis-Tris gels (Invitrogen) at 20–30 µg per well (n = 8/group), followed by separation via gel electrophoresis. Proteins were transferred onto PVDF membranes (Thermo Scientific, Waltham, MA, USA) at 75 V for two hours, followed by one hour of blocking in 1X Tris-buffered saline/Tween-20 (TBST) buffer with either 5% non-fat milk (Carnation) or 3% bovine serum albumin (BSA; Fisher Scientific, Ottawa, ON, Canada). Membranes were probed with primary antibodies overnight, followed by probing with secondary antibodies for one hour (Table 2). To avoid the excessive use of PVDF membranes, multiple targets of interest were detected on each membrane by cutting the membrane according to molecular weight and probing with the appropriate primary antibodies in separate tubes. In the case that a membrane was reused, the primary antibody was removed with stripping buffer (ThermoFisher Scientific, Waltham, MA, USA), followed by incubation with secondary antibody and re-imaging to ensure that the initial antibody was removed. Immunoreactive bands were visualized using BioRad Clarity Max Western ECL Substrate solution (Bio-Rad Laboratories Canada Ltd., Mississauga, ON, Canada) and imaged using a BioRad ChemiDoc XRS+ Imaging System. Resulting bands were analyzed using BioRad Image Lab^TM^ Software. Band intensities of target proteins were normalized to β-Actin, as previously published [45], with the exception of 4-hydroxynonenol which was normalized to total protein content detected with Ponceau staining [45,101]. Signals for proteins that were detected on the same membrane were normalized to that of β-Actin from the same membrane.

### 4.5. Statistical Analyses

At each timepoint, the selected offspring were taken from separate litters to avoid litter bias (i.e., *n* = 1 represents pups from a single dam), where a sample size of *n* = 7–8 was used for each sex per group. This sample size was chosen based on our previous studies to achieve a statistically significant difference with an expected standard deviation of 15% or less [63,102]. All statistical analyses were performed using GraphPad Prism 9 software. The results were expressed as means of normalized values ± SEM, and the threshold for significance was set as *p* < 0.05. Organ and body weight data were analyzed using Student’s two-tailed unpaired *t*-test when examining both sexes together (e.g., PND1), while sex-specific effects were analyzed by two-way ANOVA followed by a Holm-Sidak-corrected multiple comparisons test. All immunoblot data were analyzed using Student’s two-tailed unpaired *t*-test. Triglyceride and cholesterol measurements obtained from lipid analyses were analyzed by two-way ANOVA, followed by Holm-Sidak-corrected multiple comparisons test between Δ9-THC offspring and their sex-matched controls. Grubbs’ test was utilized to detect any statistical outliers.

## Figures and Tables

**Figure 1 ijms-22-07502-f001:**
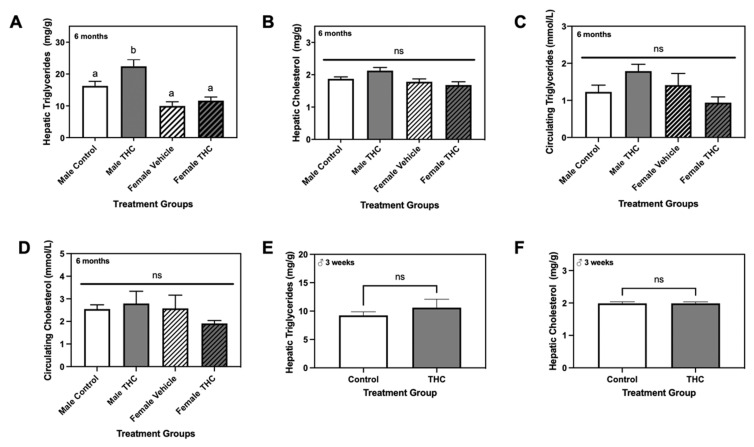
The effects of gestational Δ9-tetrahydrocannabinol (Δ9-THC) on triglyceride and cholesterol levels in the liver and plasma of exposed offspring at three weeks and six months of age. At six months of age, hepatic triglycerides (**A**) and cholesterol (**B**) were assessed in both male and female offspring (mg of lipid/g of tissue), along with circulating levels of triglycerides (**C**) and cholesterol (**D**; mmol/L). Hepatic triglycerides (**E**) and cholesterol (**F**) were also quantified in three-week old male offspring. Data are expressed as the mean ± SEM. The effects of Δ9-THC in three-week old male offspring were determined via Student’s two-tailed unpaired *t*-test, while six-month old offspring results were analyzed using a two-way ANOVA followed by a Holm-Sidak-corrected multiple comparisons test. Groups labelled with different letters are significantly different from each other (*p* < 0.05), while groups labelled ‘ns’ were non-significant from each other.

**Figure 2 ijms-22-07502-f002:**
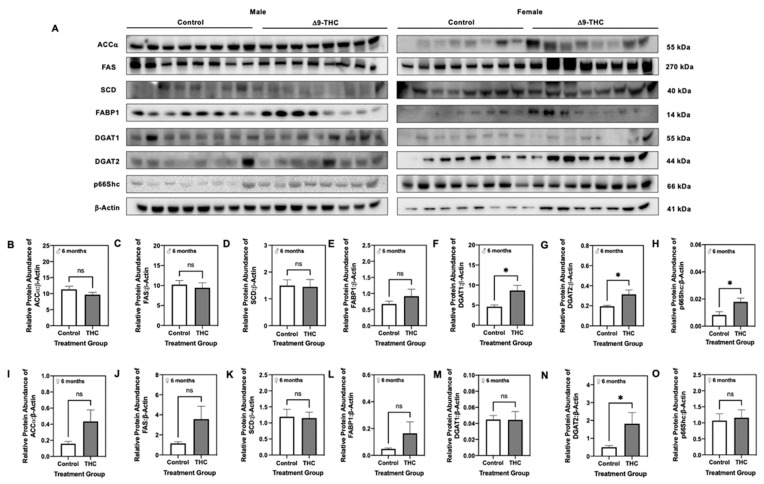
Gestational exposure to Δ9-tetrahydrocannabinol (Δ9-THC) leads to upregulation of lipogenic enzymes in male offspring at six months of age. (**A**) Representative western immunoblots illustrating hepatic expression of acetyl-coA carboxylase (ACCα), fatty acid synthase (FAS), stearoyl-coA desaturase (SCD), fatty acid binding protein 1 (FABP1), diacylglycerol acyltransferase (DGAT) 1, DGAT2, and p66Shc in male and female offspring at six months of age. Protein abundances of each enzyme for male offspring (**B**–**H**) and female offspring (**I**–**O**) were normalized to β-Actin ± SEM (*n* = 7–8/group). All protein abundances were analyzed using a two-tailed unpaired Student’s *t*-test. * Significant difference (*p* < 0.05).

**Figure 3 ijms-22-07502-f003:**
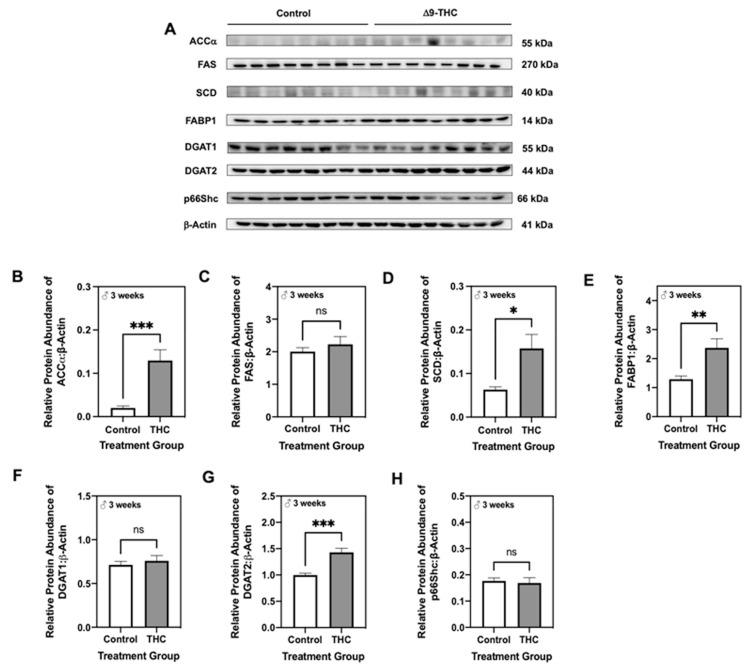
Three-week old male offspring exhibit increased lipogenic enzyme expression in the liver following gestational exposure to Δ9-tetrahydrocannabinol (Δ9-THC). (**A**) Representative western immunoblots illustrating hepatic expression of acetyl-coA carboxylase (ACCα), fatty acid synthase (FAS), stearoyl-coA desaturase (SCD), fatty acid binding protein 1 (FABP1), diacylglycerol acyltransferase (DGAT) 1, DGAT2, and p66Shc in male offspring at three weeks of age. Protein abundances of (**B**) ACCα, (**C**) FAS, (**D**) SCD, (**E**) FABP1, (**F**) DGAT1, (**G**) DGAT2 and (**H**) p66Shc were normalized to β-Actin ± SEM (*n* = 7–8/group). All protein abundances were analyzed using a two-tailed unpaired Student’s *t*-test. * Significant difference (*p* < 0.05), ** significant difference (*p* < 0.01), *** significant difference (*p* < 0.001).

**Figure 4 ijms-22-07502-f004:**
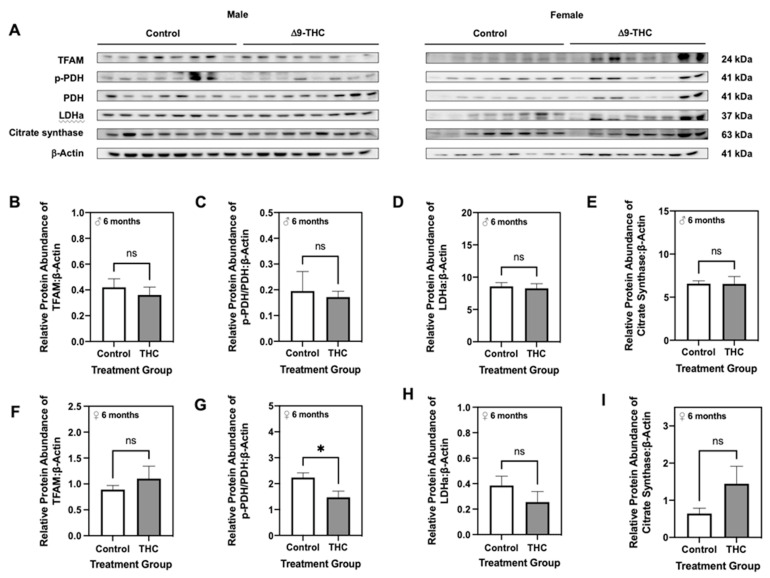
Gestational exposure to Δ9-tetrahydrocannabinol (Δ9-THC) does not impair aerobic metabolism in male or female offspring at six months of age. (**A**) Representative western immunoblots illustrating hepatic expression of mitochondrial transcription factor A (TFAM), phosphorylated pyruvate dehydrogenase (PDH) to total PDH, lactate dehydrogenase subunit A (LDHa), and citrate synthase in male and female offspring at six months of age. Protein abundances of each enzyme for male offspring (**B**–**E**) and female offspring (**F**–**I**) were normalized to β-Actin ± SEM (*n* = 7–8/group). All protein abundances were analyzed using a two-tailed unpaired Student’s *t*-test. * Significant difference (*p* < 0.05).

**Figure 5 ijms-22-07502-f005:**
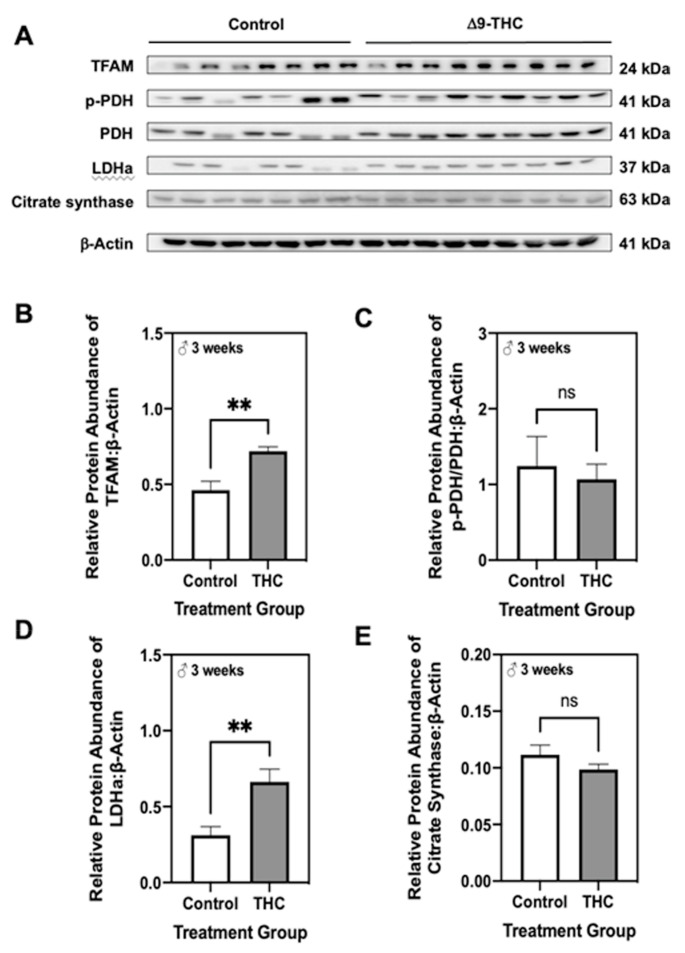
Three-week old male offspring exposed to Δ9-tetrahydrocannabinol (Δ9-THC) in utero have increased mitochondrial transcription factor A (TFAM) and lactate dehydrogenase subunit A (LDHa). (**A**) Representative western immunoblots illustrating hepatic expression of mitochondrial transcription factor A (TFAM), phosphorylated pyruvate dehydrogenase (PDH) to total PDH, lactate dehydrogenase subunit A (LDHa), and citrate synthase in male and female offspring at six months of age. Protein abundances of (**B**) TFAM, (**C**) p-PDH[Ser232]/PDH, (**D**) LDHa and (**E**) citrate synthase were normalized to β-Actin ± SEM (*n* = 7–8/group). All protein abundances were analyzed using a two-tailed unpaired Student’s *t*-test. ** Significant difference (*p* < 0.01).

**Figure 6 ijms-22-07502-f006:**
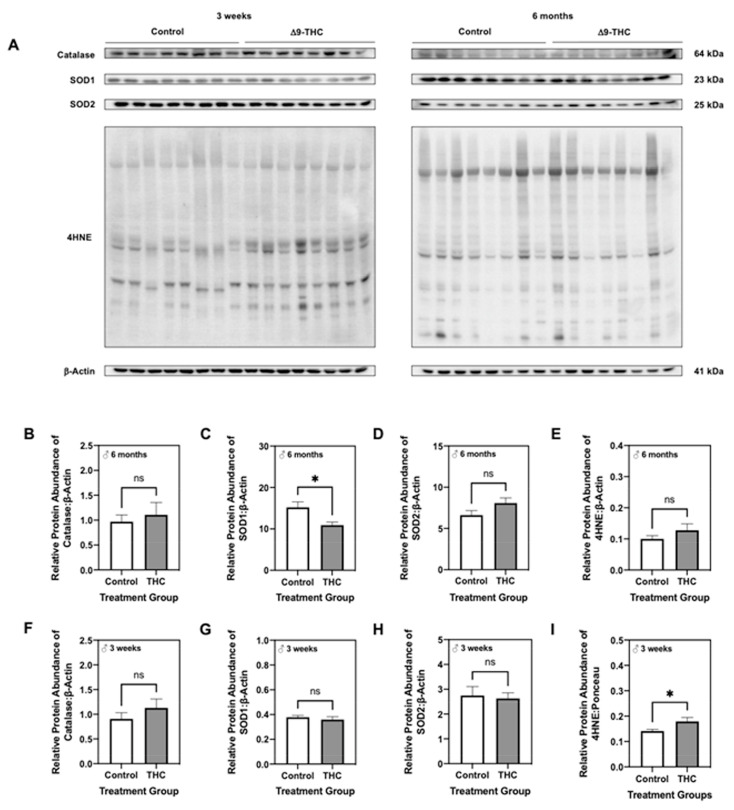
Gestational exposure to Δ9-tetrahydrocannabinol (Δ9-THC) results in hepatic oxidative stress within male offspring at three weeks and six months of age. (**A**) Representative western immunoblots illustrating hepatic expression of catalase, superoxide dismutase (SOD) 1, SOD2 and 4-hydroxynonenol (4HNE) in male offspring at three weeks and six months. Protein abundance of catalase (**B**,**F**), SOD1 (**C**,**G**), and SOD2 (**D**,**H**) were normalized to β-Actin ± SEM (*n* = 7–8/group), while 4HNE abundance (**E**,**I**) was expressed as means normalized to total protein abundance ± SEM (*n* = 7–8/group). All protein abundances were analyzed using a two-tailed unpaired Student’s *t*-test. * Significant difference (*p* < 0.05).

**Figure 7 ijms-22-07502-f007:**
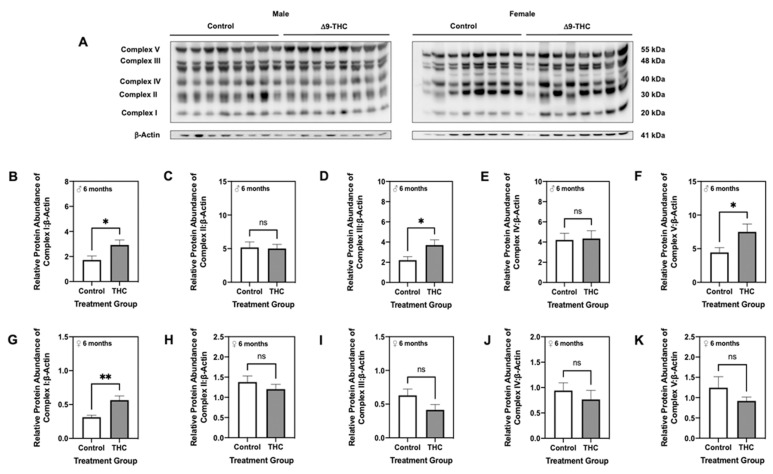
Gestational exposure to Δ9-tetrahydrocannabinol (Δ9-THC) leads to increased complexes I, III and V of the electron transport chain in male offspring at six months of age. (**A**) Representative western immunoblots illustrating hepatic expression of complexes I–V of the electron transport chain in male and female offspring at six months of age. Protein abundances of each enzyme for male offspring (**B**–**F**) and female offspring (**G**–**K**) were normalized to β-Actin ± SEM (n = 7–8/group). All protein abundances were analyzed using a two-tailed unpaired Student’s *t*-test. * Significant difference (*p* < 0.05), ** significant difference (*p* < 0.01).

**Figure 8 ijms-22-07502-f008:**
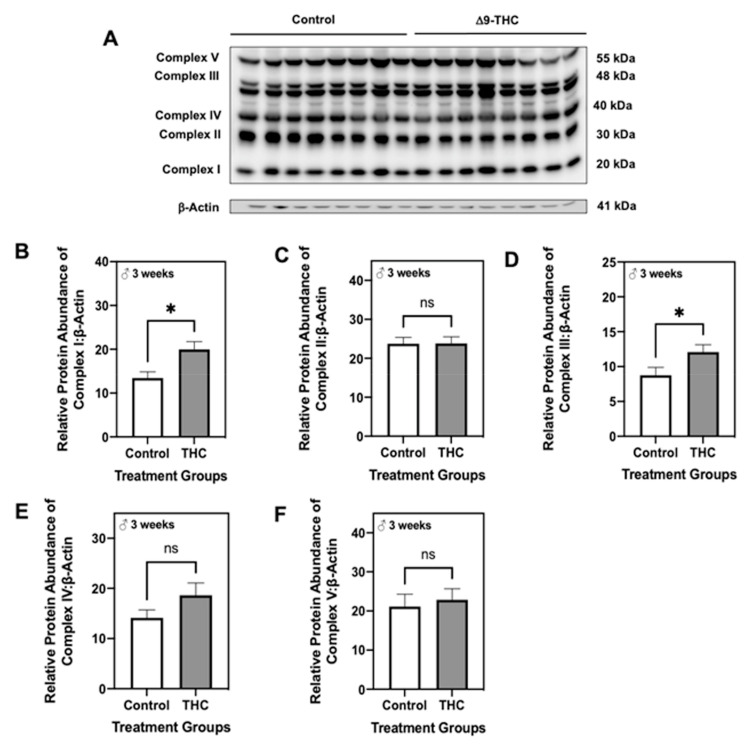
Gestational exposure to Δ9-tetrahydrocannabinol (Δ9-THC) leads to increased complexes I and III of the electron transport chain in male offspring at three weeks of age. (**A**) Representative western immunoblots illustrating hepatic expression of complexes I–V of the electron transport chain in male offspring at three weeks of age. Protein abundances of complex I (**B**), II (**C**), III (**D**), IV (**E**) and V (**F**) were normalized to β-Actin ± SEM (*n* = 7–8/group). All protein abundances were analyzed using a two-tailed unpaired Student’s *t*-test. * Significant difference *(p* < 0.05).

**Figure 9 ijms-22-07502-f009:**
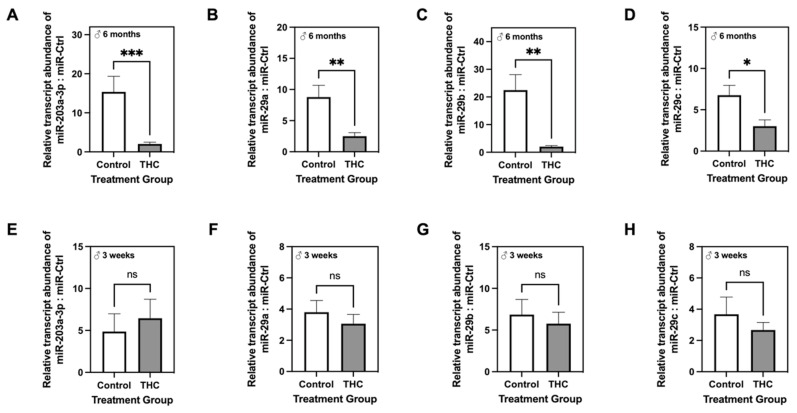
Quantitative RT-PCR analysis of miR-203a-2p and miR-29a/b/c in the livers of male offspring at six months (**A**–**D**) and three weeks (**E**–**H**) following gestational exposure to Δ9-tetrahydrocannabinol (Δ9-THC). Relative amounts of all miRs were normalized to that of miRNA standard control. Data are expressed as means ± SEM (n = 7–8/group). Groups at each time point were compared using Student’s two-tailed unpaired *t*-test. * Significant difference (*p* < 0.05), ** significant difference (*p* < 0.01), *** significant difference (*p* < 0.001).

**Figure 10 ijms-22-07502-f010:**
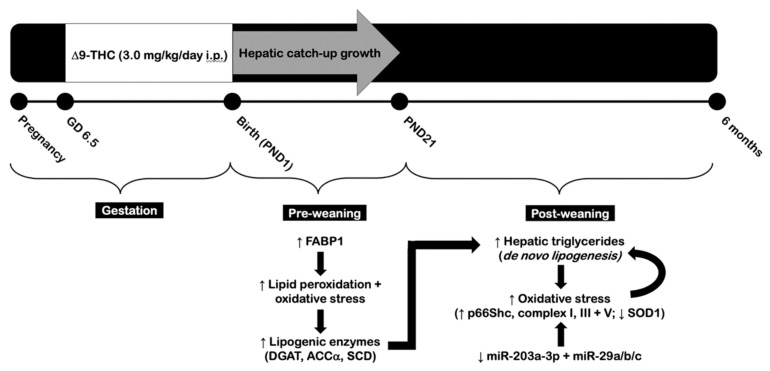
Proposed schematic illustrating the effects of gestational exposure to Δ9-tetrahydrocannabinol (Δ9-THC) on the male rat liver. In summary, in utero exposure to Δ9-THC led to symmetrical intrauterine growth restriction in the affected offspring, followed by hepatic catch-up growth and increased visceral adiposity during adult life. Male offspring exposed to gestational Δ9-THC exhibited increased FABP1 and oxidative stress at three weeks of age, leading to increased de novo lipogenesis at six months. Adult male offspring also demonstrated oxidative stress and mitochondrial dysfunction, which may occur as a result of elevated hepatic triglyceride content. The presence of oxidative stress and mitochondrial dysfunction in combination with downregulated miR-203a-3p and miR-29a/b/c may further promote impaired lipid metabolism of the liver in a positive-feedback manner.

**Table 1 ijms-22-07502-t001:** Gestational exposure to Δ9-tetrahydrocannabinol (Δ9-THC) leads to hepatic catch-up growth by three weeks and increased visceral adiposity in adult life.

	Treatment Group	Sex	Birth	Three Weeks	Six Months
Liver to Body Weight Ratio	Control	Male		0.0458 ^a^ ± 0.0026	0.0319 ^a^ ± 0.0024
		Female		0.0445 ^a^ ± 0.0017	0.0354 ^a^ ± 0.0034
		Both sexes	0.0392 ^#^ ± 0.0039	0.0451 ^#^ ± 0.0011	0.0333 ^#^ ± 0.0020
	Δ9-THC	Male		0.0459 ^a^ ± 0.0010	0.0350 ^a^ ± 0.0017
		Female		0.0437 ^a^ ± 0.0009	0.0351 ^a^ ± 0.0022
		Both sexes	0.0293 ^◆^ ± 0.0018	0.0445 ^#^± 0.0020	0.0351 ^#^ ± 0.0014
Adipose to Body Weight Ratio	Control	Male			0.01685 ^a^ ± 0.0015
		Female			0.0202 ^a^ ± 0.0025
		Both sexes			0.01828 ^#^ ± 0.0014
	Δ9-THC	Male			0.0207 ^a^ ± 0.0020
		Female			0.0234 ^a^ ± 0.0022
		Both sexes			0.0223 ^◆^ ± 0.0015

Liver growth and visceral adipose deposition were assessed for all offspring by calculating liver to body weight ratio and adipose to body weight ratio. All data are expressed as means ± SEM (*n* = 7–18/sex/group). The effects of Δ9-THC on liver to body weight ratio and visceral adipose to body weight ratio were determined via Student’s two-tailed unpaired t-test. Sex-specific differences in liver to body weight ratio were assessed using a two-way ANOVA followed by a Holm-Sidak-corrected multiple comparisons test. Groups labelled with different letters or symbols are significantly different from each other.

**Table 2 ijms-22-07502-t002:** Western blot primary and secondary antibodies, dilutions and company/catalogue information.

Antibody Name	Source	Dilution	Company (Catalogue No.)
ACCα (H-76)	Rabbit polyclonal	1:500	Santa Cruz Biotechnology Inc., Santa Cruz, CA, USA (sc-30212)
FAS (C20G5)	Rabbit monoclonal	1:1000	Cell Signaling Technology Inc., Danvers, MA, USA (#3180)
SCD (H300)	Rabbit polyclonal	1:250	Santa Cruz Biotechnology Inc., Santa Cruz, CA, USA (sc-30081)
FABP1 (D2A3X)	Rabbit monoclonal	1:1000	Cell Signaling Technology Inc., Danvers, MA, USA (#13368)
DGAT1	Rabbit polyclonal	1:1000	Novus Biologicals, Centennial, CO, USA (NB110-41487)
DGAT2 (4C1)	Mouse monoclonal	1:1000	Santa Cruz Biotechnology Inc., Santa Cruz, CA, USA (sc-293211)
SHC	Mouse monoclonal	1:1000	BD BioSciences, San Jose, CA, USA (610879)
TFAM (D5C8)	Rabbit monoclonal	1:1000	Cell Signaling Technology Inc., Danvers, MA, USA (8076)
pSer(232) pyruvate dehydrogenase	Rabbit polyclonal	1:1000	EMD Millipore, Etobicoke, ON, Canada (AP1063)
Pyruvate dehydrogenase	Rabbit polyclonal	1:1000	Cell Signaling Technology Inc., Danvers, MA, USA (2784)
LDHa	Rabbit polyclonal	1:1000	Cell Signaling Technology Inc., Danvers, MA, USA (2012)
Citrate Synthase	Rabbit polyclonal	1:1000	Provided by Dr. S. Raha, McMaster University
Catalase (H-300)	Rabbit polyclonal	1:1000	Santa Cruz Biotechnology Inc., Santa Cruz, CA, USA (sc-50508)
Superoxide dismutase (SOD)-1 (FL-154)	Rabbit polyclonal	1:1000	Santa Cruz Biotechnology Inc., Santa Cruz, CA, USA (sc-11407)
Superoxide dismutase (SOD)-2 (FL-222)	Rabbit polyclonal	1:1000	Santa Cruz Biotechnology Inc., Santa Cruz, CA, USA (sc-30080)
4-hydroxynonenal	Mouse monoclonal	1:1000	R&D Systems, Oakville, ON, Canada (MAB3249)
OXPHOS rodent cocktail	Mouse monoclonal	1:1000	Abcam Inc., Toronto, ON, Canada (ab110413)
β-Actin Peroxidase	Mouse monoclonal	1:25,000	Sigma Aldrich Co., St. Louis, MO, USA (A3854-200UL)
Goat anti-rabbit IgG HRP-linked (H+L chain)	N/A	1:10,000	Cell Signaling Technology Inc., Danvers, MA, USA (7074P2)
Horse anti-mouse IgG HRP-linked (H+L chain)	N/A	1:10,000	Cell Signaling Technology Inc., Danvers, MA, USA (7076S)

## Data Availability

Not applicable.

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
