# Peer review of "In Utero Exposure to Δ9-Tetrahydrocannabinol Leads to Postnatal Catch-Up Growth and Dysmetabolism in the Adult Rat Liver"

_ijms, 2021, doi:10.3390/ijms22147502_

Round 1
Reviewer 1 Report
The submitted reseach article analyses the effects of in utero exposure to the phytocannabinoid THC on liver catch-up growth and lipid metabolism in the offspring. In general the manuscript is well structured and organized, data support conclusion. Possible epigenetic mechanisms involving the differential expression of specific miRNAs is also proposed.
My only query concerns Western blot. Provided figures are quite small, in few cases duplicated (aspecific?) bands seems to be cut from the original blot and the quality of Western blot for ACCα (figure 3), SCD (figures 2-3), tfam (figure 4), catalase (figure 7) appears quite poor. May the authors provide uncut original figures with the indication of the specific quantifyed bands? Furthermore in figure 5 the signals for p-pdh, pdh and LDHa appears to be of different sizes in control group.
Lastly beta-actin panels seems to be the same in different figures.
What about the expression of cannabinoid receptor in the liver of treated animals?
Author Response
Point 1: My only query concerns Western blot. Provided figures are quite small, in few cases duplicated (aspecific?) bands seems to be cut from the original blot and the quality of Western blot for ACCα (figure 3), SCD (figures 2-3), tfam (figure 4), catalase (figure 7) appears quite poor. May the authors provide uncut original figures with the indication of the specific quantifyed bands? Furthermore in figure 5 the signals for p-pdh, pdh and LDHa appears to be of different sizes in control group.
Response 1: Thank you for these insightful comments, we understand where the concern arises. First, we have modified Figure 2 such that the western blot images are larger and easier to see. Since our study involved the relative quantification of numerous proteins, some markers of interest were targeted on the same PVDF membrane. This involved cutting the membrane based on protein molecular weight; therefore, some images do not include the entire membrane. That said, we have provided the original images for each target and included them as supplementary figures in the manuscript. We have also indicated the row of bands that was analyzed with an arrow on each image. Note that for some proteins, tape was used to cover non-specific bands that were of higher abundance and therefore stronger density. We have found that these bands take away from the signal from the protein of interest, while covering up these stronger bands helps to enhance the signal of the specific target (i.e., as in the case of SCD).
Point 2: Lastly beta-actin panels seems to be the same in different figures.
Response 2: This is an important point. As mentioned above, PVDF membranes were cut based on the molecular weight of the proteins of interest, followed by incubation in separate tubes with the respective primary antibodies. In the case that a membrane was reused to quantify another protein, the membrane was stripped with stripping buffer for 5 minutes, re-incubated with secondary antibody, and re-imaged to ensure that the initial primary antibody was completely stripped. Proteins that were detected on the same membrane were normalized to the relative abundance of b-Actin for that particular membrane. While not all protein markers in a given figure were detected on the same membrane, the b-Actin images in each figure are representative to avoid having multiple blots shown for b-Actin. We have now added statements to clarify this in the “Materials and Methods” section of the manuscript (lines 584–590 and 595–596).
Point 3: What about the expression of cannabinoid receptor in the liver of treated animals?
Response 3: Thank you for this suggestion. While it is possible that there may be changes in cannabinoid receptor expression, we did not examine this in the current study. Discrepancies in the expression of CB1R and/or CB2R could absolutely occur upstream of the observed effects. This is something that we will investigate further in the future, possibly in studying THC pharmacokinetics and pharmacodynamics within these offspring. We have now indicated that this is a limitation of the current study in the discussion section of the manuscript (lines 484–488).
Reviewer 2 Report
This study analyzes the role of gestational exposure to tetrahydrocannabinol on the development and function of the liver with dyslipidemia due to mitochondrial dysfunction and epigenetic mechanisms.
It is a well-written manuscript, presenting all the information needed for a good understanding of the field, the study (in a logical way of all steps), discussions, and conclusions. The results are supported by many figures for a better understanding. Also, in discussions, the authors proposed a scheme of the effects of the gestational exposure to THC on the male rat liver.
I have no recommendations to make for improving the paper.
Author Response
Responses to Reviewer 2
Point 1: It is a well-written manuscript, presenting all the information needed for a good understanding of the field, the study (in a logical way of all steps), discussions, and conclusions. The results are supported by many figures for a better understanding. Also, in discussions, the authors proposed a scheme of the effects of the gestational exposure to THC on the male rat liver. I have no recommendations to make for improving the paper.
Response 1: We thank the reviewer for all of their positive comments.